# Understanding Communication Barriers: Demographic Variables and Language Needs in the Interaction between English-Speaking Animal Professionals and Spanish-Speaking Animal Caretakers

**DOI:** 10.3390/ani14040624

**Published:** 2024-02-15

**Authors:** Allen Jimena Martinez Aguiriano, Leonor Salazar, Silvana Pietrosemoli, Marcelo Schmidt, Babafela Awosile, Arlene Garcia

**Affiliations:** 1School of Veterinary Medicine, Texas Tech University, Amarillo, TX 79106, USA; allemart@ttu.edu (A.J.M.A.); leonor.salazar@ttu.edu (L.S.); marcelo.schmidt@ttu.edu (M.S.); babafela.awosile@ttu.edu (B.A.); 2Department of Animal Science, College of Agriculture and Life Sciences, North Carolina State University, Raleigh, NC 27695, USA

**Keywords:** language needs, animal welfare, Spanish for Specific Purposes in Agriculture, Hispanic workforce

## Abstract

**Simple Summary:**

This study aimed to understand the language challenges faced by English-speaking animal professionals when communicating with Hispanic/Spanish-speaking animal caretakers. A survey was conducted among bilingual and non-Spanish-speaking professionals, revealing that non-Spanish-speaking individuals struggled with both written and oral communication compared to bilingual counterparts. Female professionals exhibited differences in their responses, particularly regarding the importance of certain aspects of the Spanish language while interacting with Hispanic caretakers. These findings highlight communication gaps that need to be addressed to improve interactions with on-farm Hispanic/Spanish-speaking animal caretakers and consequently contribute to enhancing animal health, welfare, and production.

**Abstract:**

This study focused on assessing the language needs of English-speaking animal professionals in their interactions with Hispanic/Spanish-speaking animal caretakers. A survey was administered to a target audience of non-Spanish speaking and bilingual animal professionals to identify communication gaps while interacting with Hispanic/Spanish-speaking animal caretakers. The data was analyzed with descriptive and inferential statistics, including ordinal regression analyses to examine the impact of demographic variables on respondents’ answer choices. The results showed that English-speaking professionals struggled with written and oral communication, which differed compared to bilingual professionals (*p* < 0.05). Additionally, responses of female professionals varied regarding the aspects of Spanish necessary for interacting with Hispanic/Spanish-speaking animal caretakers, as well as the topics likely to be addressed when agriculture professionals communicate with animal caretakers (*p* < 0.05). Communication difficulties in the oral medium for both oral receptive skills (listening) and oral productive skills (speaking) were reported as the major barriers that animal professionals need to overcome in their attempt to communicate with the Hispanic/Spanish-speaking workforce in farm settings. This emphasizes the need to address oral communication barriers, and to a lesser degree, the development of reading and writing skills. The topics: typical clinical signs of illness, euthanasia, treatment—drugs, and identification of sick or injured animals were identified as the most likely to be addressed during on-farm interactions. These findings indicate that there are gaps in communication that need to be overcome to improve communication with on-farm Hispanic/Spanish-speaking animal caretakers and consequently contribute to enhancing animal health, welfare, and production.

## 1. Introduction

The United States is experiencing a rise in the number of Spanish-speaking immigrants with limited English proficiency [1]. Currently comprising 18.5% of the U.S. population (over 60 million individuals), Hispanics are the largest minority group in the U.S. [2]. About 17.2 million (43%) of Hispanics are mono-lingual Spanish speakers [3]. Projections suggest that this group will represent 30% of the U.S. population by 2050 [4]. Notably, the number of Hispanic farm workers in the U.S. has been increasing at a faster rate than the general population [5]. This trend can be attributed to the physically demanding nature of farming work, which requires specialized skills and knowledge. Many Hispanic workers have prior experience and have acquired their farming abilities in their home countries, where wages are often lower than in the U.S. A study conducted by the National Center for Farmworker Health [6] revealed that the majority (62%) of agricultural workers reported that they were most comfortable conversing in Spanish; 29% said they could not speak English “at all”, 39% said they could speak “somewhat/a little bit” of English.

This article addresses the first stage of a broader research project funded by the U.S. Department of Agriculture (USDA). While this work aimed at identifying the gap in communication between animal professionals and the Hispanic workforce, the specific goal of the entire project is to utilize the findings to develop and implement three courses of Spanish for Specific Purposes in Agriculture (SSPA) that will be addressed to animal science and veterinary medicine students. The objective of this specific study is to analyze and understand the communication needs that non-bilingual animal professionals have during their work interaction with Hispanic/Spansish-speaking animal caretakers. A thorough needs analysis was carried out to identify the communication requirements of this demographic group. The results will be used to address the communication gaps between animal professionals and the Hispanic workforce. Our work can be used as a model for other industries experiencing language barriers.

### 1.1. Communication Gap

Spanish for Specific Purposes (SSP) is an area of study that specializes in teaching Spanish for specific contexts. This approach has gained interest in the agricultural industry, where it has been used to promote the development of language skills for communication in specialized fields [7,8] such as animal welfare. In this particular field, the successful implementation of standards, for instance, depends on the key contribution of farm workers [9]. Effective communication among farmers, animal-welfare organizations, veterinary/animal-science experts, and farm workers is essential in ensuring animal-welfare practices, livestock productivity, profitability, and sustainability. In the U.S. livestock-farming industry, Spanish language skills are particularly valuable among non-bilingual professionals, as a sizable portion of the workforce speaks Spanish as their first language and has limited proficiency in English [9].

Several studies [10] have suggested that communication with animal-production workers who speak a language other than English at home involves the necessity to translate any content into their primary language. However, the communication strategies targeting diverse cultural groups are not well understood yet. 

The inability to communicate can affect the exchange of information, resulting in potential negative outcomes for those involved. SSP seems to be an effective tool to equip English speakers with the necessary Spanish vocabulary, grammar structures, and communication skills, to help them effectively interact in their specific work areas. Research has demonstrated the effectiveness of SSP in the acquisition of specific language skills for the workplace, particularly in the medical field [11]. 

Prior research experiences deal with SSP in veterinary medicine. Colorado State University (CSU) has developed a program called “Spanish for Veterinarians Language Program” (SVLP) to bridge the Spanish–English language gap in this professional context. This initiative included a pilot delivery in 2021 and a second one in 2022 [12]. These efforts focused on supplying the necessary skills for successful communication concerning animal health. They also intended to reinforce the importance of mastering the Spanish language, as Spanish is the second-most spoken language in the U.S. [13]. Results derived from a survey administered to veterinary students from CSU revealed that 93.5% of the students are interested in learning basic Spanish language skills to overcome professional needs they might experience in the future [14].

The effective delivery of animal and veterinary programs requires cultural competence in professional and research practice, considering that veterinarians often work with culturally and linguistically diverse teams [15,16]. Therefore, addressing language gaps could potentially help to target cultural background differences. Research has proved that SSP is an effective teaching method for acquiring specific language skills for the workplace [7,17]. The use of SSP in agricultural (AG) education is fundamental in ensuring that workers develop the necessary communication skills to discuss topics on animal health, welfare, production, and food safety. Numerous disciplines have benefited from implemeting SSP [18,19,20]. It has been established that SSP enhances language-learning outcomes and improves workplace communication [15,21]. Some managers of Hispanic workers in the horticulture industry in Iowa have expressed a need for educational programs to bridge language and cultural gaps [22].

To address the language barrier between Spanish-speaking workers and English-speaking managers, various agricultural agencies and land grant universities have developed culturally sensitive education, translation, and training tools for farm owners on how to interact effectively with the Hispanic workforce. These resources for enhancing management practices are a response to the acknowledged demographic shift in the U.S. workforce [23].

Demographic factors, including language proficiency and educational background, can play a significant role in determining the effectiveness of communication between agriculture professionals and Hispanic animal caretakers. Research on this topic has revealed that language barriers often impede communication, resulting in misunderstandings, inefficiencies, and potentially dangerous practices in animal care. It is essential to target these demographic factors to enhance communication in the agriculture industry [24].

### 1.2. Needs Analysis and Language for Specific Purposes

Needs analysis (NA) is a well-known fundamental concept in language teaching and syllabus design. Although different concepts of needs analysis exist, it may be defined as “a systematic process of inquiring about the communicative language needs of learners prior to the design of a syllabus or a study program to provide appropriate goals, learning materials, and strategies in a constructive learning environment” [25]. By gathering information from the field where activities are conducted in the “real world”, a NA is used to establish how the language course will be conceived and delivered, and what content should be included. This process has been considered to be the first step in developing courses of languages for specific purposes (LSP) since it will ensure that students engage in activities conducive to learning the topics and developing the language skills that they need to successfully perform in academic or occupational settings [26].The analysis of the target situation determines the communicative needs of learners; that is, what they need to know to function effectively in that specific language context. Needs analysis helps course designers define the needs of learners as accurately as possible to meet their goals for real language use in their field of work or study.

According to West [27], the term needs analysis was first introduced by Michael West in India in the 1920s and was later adopted in language curriculum design. With the advent and emergence of English for Specific Purposes and through the contribution of applied linguists and educational researchers [28,29,30,31,32], the concept was thoroughly researched and became well known within the theoretical framework and practice of Communicative Language Teaching (CLT). As a result, needs analysis has long been considered a prerequisite in the design of courses of languages for specific purposes.

Trim [33] was decisive in developing functional notional models for specifying language learning objectives based on the purpose of communication (functions) and general concepts or categories (notions). In later years, different approaches, methods, and techniques to carry out a needs analysis were conscientiously devised for syllabus design and curriculum development through the work of many authors and researchers, who emphasized its relevance prior to course design.

Hutchinson and Waters [32] for instance, stressed that needs analysis is a crucial step in course design, while pointing out the difference between needs (the communicative demands in the target situation), wants (the desires and expectations of the learner), and lacks (the gap between what the learner already knows and the desired ability). Brindley [25] emphasized the difference between objective and subjective language needs: the former includes information about the learners themselves, and the latter relates to their cognitive and affective factors. Brindley also pointed out three different views of needs: what learners need to do in each target situation (goal-oriented needs), what they need to do to learn (process-oriented needs), and those derived from the specific goal or target situation (product-oriented needs).

However, it is necessary to highlight that the importance of needs analysis extends beyond syllabus design. As Knox [34] states, it is useful for teachers as well as for learners and institutional administrators during all stages of the program. Initially, it was used to design the program, course contents, and materials; during the program, it ensures the achievement of goals and provides opportunities for necessary changes; and finally, after the program has been implemented, it is useful for assessment and evaluation to devise changes and plans for future directions.

The change from teacher-centered approaches to student-centered approaches in language teaching was a major factor in the promotion and implementation of needs analyses. Teachers, administrators, and material designers need to find the characteristics and needs of the students in advance to design courses and materials tailored to the learners’ communicative needs in the target situation. In this context, teaching methods and strategies changed by the new approaches and theoretical developments; authentic materials were chosen according to the relevant topics and types of discourse suitable to the context and field of study or area of work of the participants; communicative activities were developed based on the concept of an information gap, pair work, and group work; and specific skills were targeted according to the type of course: reading, writing, listening, or speaking. The development of student learning-autonomy has become a primary goal when product-oriented approaches evolve into process-oriented approaches that consider different learning strategies and styles of students. In this changing educational context, the use of a needs analysis had, and still has, a key role to play in defining achievable learning outcomes.

Throughout the years, different approaches to need analysis have been developed, and several methods have evolved to carry it out, the most common being field surveys via questionnaires and/or structured interviews. These are designed in such a way that information can be gathered on both the target needs and the learning needs of participants, including the characteristics, educational background, needs, and expectations, as well as the lack of prospective language students. The questionnaires and/or interviews can be administered to key informants: learners, administrators, teachers, and/or professionals in a specific field or discipline. Nowadays, with the help of technology, surveys and questionnaires can be built and answered online using available open-source survey software. In addition to surveying through a questionnaire, the analysis of authentic texts and samples of the target language is another useful technique to decide the key features of discourse to register in the target context to choose proper materials and the skills to handle them.

### 1.3. Present Tendencies: The Intercultural Dimension

After more than fifty years of research and practice, the use of needs analysis is not yet outdated, and it is still considered relevant today in strategic planning and program evaluation. In recent years, the preferred term has been Needs Assessment, and an interesting turn in perspective has taken place: the relevance assigned to the cultural part of language ability.

Since the 1990s, constructivist approaches to language teaching and the development of pragmatics as an important subfield of linguistics have been highly relevant to the intercultural dimension and have promoted increased awareness of the cultural domain of linguistic competence. This is called intercultural competence, defined as “the ability to negotiate and mediate between multiple identities and cultures” [35,36] and it means that the foreign language learner/speaker should be able to master the dynamics of intercultural communication as part of their competence in the target language.

According to Deardorff [37], intercultural competence includes three common dimensions: cognitive (what the student knows), affective (what the student understands and values), and behavioral (what the student can do). These dimensions have been included as part of the descriptors of language competence in the National Standards for Foreign Language Education [38], the Common European Framework of Reference for Languages [39], and language programs worldwide [35,40]. The three dimensions of intercultural competence (i.e., knowledge, attitudes, and skills) interact with other key elements to account for the degree of success of language curricula and are therefore relevant in defining the learning context as well as the learner’s needs.

Based on the information above, a needs analysis was performed to analyze and understand the communication needs that English-speaking animal professionals have during their work interaction with Hispanic/Spanish-speaking animal caretakers.

## 2. Materials and Methods

This study was approved by The Human Research Protection Program at Texas Tech University (IRB2021-250). The research involved the design and administration of an anonymous online survey to English-speaking and bilingual animal professionals. The survey was the result of a joint effort of specialists in the fields of animal production and linguistics with the purpose of ensuring that both agricultural aspects and linguistic components were addressed properly. The survey was administered to animal professionals including: veterinarians, animal scientists, farm owners, farm managers, animal nutritionists, agribusiness consultants, farm trainers, and professors, who complied with the criterion of having prior communication experience with Spanish-speaking animal caretakers. The aim of this survey was to identify the language needs of these professionals when interacting with Hispanic/Spanish-speaking animal caretakers.

### 2.1. The Data Collection Instrument 

Established animal welfare, animal health, and food safety standards were thoroughly reviewed to identify key topics and vocabulary to be included among the survey items. The survey focused on specific livestock species: bovine (dairy/beef cattle), swine, and poultry (layers, broilers and turkeys). This selection of the species was the result of having considered the economic relevance of such species within the geographical location of the three universities involved in the research project, Texas Tech University, North Carolina State University, and Tarleton State University.

The survey comprised three sections: Section A focused on the demographic information of the participants, while Sections B I and B II examined participants’ language needs and functional use of the language in farm settings. A five-point Likert scale ranging from Strongly disagree (=0 points) to Strongly agree (=5 points) was used to assess respondents’ degree of agreement on the proposed options both in sections B I and B II. The preliminary version of the instrument was validated by 11 national and international experts in linguistics and their suggestions were incorporated into the final version administered to the sample (Appendix A).

### 2.2. Descriptions of Sections of the Survey

**Section A** dealt with demographic information distributed in eight items addressing gender, age, race, ethnicity, proficiency in languages, profession/occupation, education/degree, and expertise with species.

**Section B I** included five statements to explore the major issues faced in on-farm activities while interacting with Hispanic/Spanish-speaking animal caretakers and the respondents’ language needs and functional use of the Spanish language.
**1.** **The major issues faced when trying to interact with Spanish-speaking animal caretakers on-farm are …** (4 answer choices);**2.** **Veterinarians, animal scientists, and other professionals need Spanish in on-farm activities in order to …** (4 answer choices);**3.** **Agriculture professionals need to communicate verbally with animal caretakers mainly to …**(13 answer choices);**4.** **For the development of on-farm activities, it is important that veterinarians, animal scientists, and other animal professionals communicate in written language (Spanish) to …**(8 answer choices);**5.** **These aspects of the Spanish language are deemed necessary for interacting with Spanish-speaking animal caretakers …** (6 answer choices).

This section helped identify perceived needs of Spanish language skills in relation to various on-farm tasks. Furthermore, the survey explored the specific aspects (grammar, vocabulary, listening, speaking, reading, and writing) of the Spanish language considered essential for effective communication. This group of questions formed a robust foundation to understand language-related challenges in on-farm interactions with Hispanic/Spanish-speaking animal caretakers. 

**Section B II** explored the likelihood of addressing specific topics when agriculture professionals communicate with Hispanic/Spanish-speaking animal caretakers. The statement was “The following topics are likely to be addressed when agriculture professionals communicate with animal caretakers”. The answer choices for this statement were animal handling, pain management, husbandry practices, proper identification of sick animals, typical signs of illness, abnormal behavior patterns, animal behavior and emotional states, antimicrobial resistance, biosecurity, birthing, body conditions, environmental conditions, euthanasia, feeding, housing, hygiene, parasites, predators, record keeping, traceability, treatment—drugs, vermin control, and worker health.

### 2.3. Sampling Procedure

A flier was distributed at a conference inviting potential participants to take part in the survey. To participate, respondents were required to access a link to the online survey and follow the instructions provided. The sample size was not predetermined. It was a convenience sample, trying to involve as many willing individuals as possible to participate. A total of 39 individuals took the survey. Although this sample was small, this study can be considered a pilot or foundational study and can be used as a departing point for a larger study, since respondents were part of a limited group of professions comprising veterinarians, animal scientists, and others (farm owners, farm managers, animal nutritionists, agribusiness consultants, farm trainers and professors).

It is crucial to underscore that this study is novel for this geographical area and specific professions. Previous research has not delved into the influence of demographic variables on the perception of the need to understand or learn another language within this context.

To ensure the integrity of the data collected, quality control measures were implemented. These included the anonymity of the respondents and the survey was available for a limited time (August 2021–October 2021) to minimize the potential for response bias.

The research questions focused on communication issues faced by veterinarians, animal scientists, and other animal professionals when interacting with Hispanic/Spanish-speaking animal caretakers. Respondents were asked to answer a series of questions, and their responses were collected via the anonymous online survey. 

### 2.4. Data Analysis

The survey was conducted using the Survey Monkey platform. Once the participants completed the survey, the information was downloaded, organized, and managed in Microsoft Excel® for Microsoft 365 MSO (Version 240).Once organized, it was imported into R statistical software (Version 4.3.2) which facilitated the exploration of patterns and trends in the responses. Descriptive statistics were used to summarize the data and their percentages with 95% confidence intervals (CIs). Inferential statistics were also performed, specifically ordinal regression analyses, to test the hypotheses set for this study: demographic variables do not affect the answer choices of the respondents.

The purpose for using this statistical model was to achieve a deeper understanding of the ordinal nature of the variables and their interactions with other factors. To quantify the strength and direction of these associations, odds ratios (ORs) were computed for each of the variables, with 95% CI. The calculation of ORs and CI added precision to the variables analysis, providing a deeper interpretation of the impact of explanatory variables on the response variables. Ordered ORs > 1 showed odds or tendency towards strong agreement versus the combined adjacent response categories, given that other variables are held constant in the model. An OR < 1 showed odds or tendency towards strong disagreement versus the combined adjacent response categories, given that other variables are held constant in the model.

## 3. Results

### 3.1. Demographic Survey Frequencies

A total of 54 individuals started the survey but only 39 finished it, for a completion rate of 72.2%. Most participants identified as male (69.23%, n = 27), while 28.21% (n = 11) identified as female. One participant did not answer this question (2.56% n = 1). The racial distribution of the sample was predominantly white (92.31%, n = 36), and the remaining respondents did not specify their race (7.69%, n = 3). Most participants identified as non-Hispanic/Latino (71.79%, n = 28), while only 23.08% (n = 9) were Hispanic/Latino, and the remaining respondents did not specify their ethnicity (5.13%, n = 2). In terms of language proficiency, most participants reported English as their primary language (61.54%, n = 24), while another proportion reported proficiency in both English and Spanish (33.33% n = 13), and a smaller proportion reported proficiency just in the Spanish language (5.13%, n = 2).

Regarding profession/occupation, the sample was diverse with the following proportions: veterinarians (20.51%, n = 8), animal scientists (20.51%, n = 8), farm owners (5.13%, n = 2), and farm managers (7.69%, n = 3), while 46.15.% (n = 18) identified as having other professions or occupations such as animal nutritionists, agribusiness consultants, farm trainers, and professors.

In terms of education level, most participants had a bachelor’s degree (41.03%, n = 16) or a DVM (Doctor of Veterinary Medicine) degree (20.51%, n = 8). Another proportion had a master’s degree (20.51%, n = 18) or a PhD (10.26%, n = 4) degree, while 3.51% and 7.02% had an associate degree and other levels of education, respectively. Prior experience working with animal species varied among participants, with the largest proportion reporting to have worked with swine (53.85%, n = 21) and cattle (41.03%, n = 16).

### 3.2. The Impact of Demographic Factors on Communication between Agriculture Professionals and Hispanic/Spanish-Speaking Animal Caretakers

#### 3.2.1. Comparisons Based on the Language That Respondents Are Proficient in

Different comparisons were performed between non-bilingual professionals (only English-speaking) and bilingual (English and Spanish) professionals for specific questions posed and the responses provided. There were no major statistical differences for most of the comparisons. However, when compared to the bilingual-speaking professionals, the non-bilingual professionals demonstrated significant odds or a tendency towards strong agreement in their responses to the item “Major issues faced when trying to interact with Spanish-speaking animal caretakers on farm” (Figure 1). Only non-bilingual professionals strongly agreed that there is “difficulty in understanding what is written in Spanish by the workers”compared to the bilingual-speaking professionals (odds ratio: 54, 95% CI: 10–403, *p* = 0.001). Additionally, compared to the bilingual-speaking professionals, the non-bilingual professionals showed a tendency towards strong agreement with the statement “Difficulty to understand what they tell me“ (odds ratio: 13.2 CI: 3.24–65.5, *p* = 0.0006). This tendency towards agreement was also perceived for the questions “I don’t feel confident writing things in Spanish” (odds ratio: 54.76 CI 10–46 *p* = 0.00023) and “I have a hard time trying to express my thoughts in words” (odds ratio: 5.22 CI 1.4–20.1 *p* = 0.012). Other results for headlines can be found in the Appendix A.

#### 3.2.2. Comparisons Based on Gender

In terms of gender, different comparisons were performed between male and female professionals for the specific questions posed and the responses provided. There were no major statistical differences for most of the comparisons. However, when compared to male professionals, female professionals showed statistically significant odds or tendency towards strong agreement in their responses to the statement “Aspects of the Spanish language deemed necessary for interacting with Spanish-speaking animal caretakers” (Figure 2).

Female professionals exhibited strong agreement in agriculture vocabulary (odds ratio: 0.067 CI: 0.003–0.42 *p* = 0.001), listening comprehension (odds ratio: 0.18 CI: 0.3–0.77 *p* = 0.029) and speaking (odds ratio: 0.20 CI: 0.03–0.88 *p* = 0.043) as the main topics. Furthermore, female professionals showed statistically significant odds or tendency towards strong agreement in their responses to the statement on “Topics likely to be addressed when agriculture professionals communicate with animal caretakers” (Figure 3). Female professionals strongly agreed on euthanasia (odds ratio: 0.10 CI: 0.005–0.66 *p* = 0.04) and parasites (odds ratio: 4.23 CI: 1.18–16.51 *p* = 0.03) as the main topics that should be addressed, compared to male professionals. Other results for headlines can be found in the Appendix A.

#### 3.2.3. Comparisons Based on Ethnicity (Hispanic vs. Non-Hispanic)

In terms of ethnicity, different comparisons were made between Hispanic and non-Hispanic professionals for the specific questions posed and the responses provided. There were no major statistical differences for most of the comparisons. However, when compared to Hispanic professionals, non-Hispanic professionals displayed statistically significant odds or a tendency towards strong agreement in their responses to the question posed on the importance of communicating in written language (Spanish) for the development of on-farm activities (Figure 4). Hispanic professionals strongly agreed that it is necessary to provide written guidelines for animal genetic improvement and breeding programs compared to non-Hispanic professionals (odds ratio: 0.16, 95% CI: 0.03–0.74, *p* = 0.022). Additionally, Hispanic professionals showed a tendency towards strong agreement with the statements “Understand animal records written by caretakers” (odds ratio: 0.20, CI: 0.03–0.89, *p* = 0.0004) and “Write instructions about feeding programs” (odds ratio: 0.18, CI: 0.03–0.89, *p* = 0.004).

The results of the statistical analysis suggest that Hispanic professionals showed a significant inclination towards strong disagreement with respect to the major issues encountered when interacting with Hisapnic/Spanish-speaking animal caretakers (Figure 5). They exhibited a strong disagreement with the notion that there is difficulty in understanding written information provided by animal caretakers in Spanish compared to their non-Hispanic counterparts (odds ratio: 28.2, 95% CI: 5–205, *p* = 0.00002). Additionally, Hispanic professionals displayed a propensity towards strong disagreement with the statements “Difficult to understand what they tell me” (odds ratio: 29.8 CI: 5.8–204.3, *p* = 0.00001), “I don’t feel confident writing things in Spanish” (odds ratio: 24.5 CI: 5.03–153.5, *p* = 0.0001), and “I have a hard time trying to express my thoughts in words” (odds ratio: 16.3 CI: 3.51–93.8, *p* = 0.0007).

On the other hand, Hispanic professionals showed concurrence with non-Hispanic professionals in refernce to the main skills necessary for interacting with Hispanic/Spanish-speaking animal caretakers. Specifically, they demonstrated strong agreement with the importance of “Reading” (odds ratio: 0.15 CI 0.02–0.75 *p* = 0.027) and “Writing” (odds ratio: 0.12 CI 0.02–0.58.11 *p* = 0.012). The data revealed that Hispanic professionals demonstrated a tendency towards strongly agreeing with the statement “Topics likely to be addressed when agriculture professionals communicate with animal caretakers,” especially regarding the topic “Parasites” (odds ratio: 0.22, confidence interval [CI] 0.04–0.95, *p* = 0.054), as illustrated in Figure 6.

#### 3.2.4. Comparisons Based on Profession/Occupation

No statistically significant differences were found among animal professionals (veterinarians, animal scientists, farm owners, farm managers, animal nutritionists, agribusiness consultants, farm trainers, and professors) regarding the specific questions posed and the corresponding responses.

#### 3.2.5. Comparisons Based on Work Experience with Specific Animal Species

There were no statistical differences between animal professionals working with different animal species (cattle, swine, poultry, or with experience in multispecies) with respect to the specific questions asked and the responses provided.

### 3.3. Language Needs

The following sections reflect each of the items and subitems included in Parts B I and B II of the survey. These sections are devoted to exploring the language needs and functional use of Spanish. The results are presented with their corresponding percentages and 95% confidence intervals (CIs).

#### 3.3.1. Major Issues Faced When Trying to Interact with Hispanic/Spanish-Speaking Animal Caretakers Caretakers on Farm Settings

The fundamental issues faced by non-bilingual animal professionals when interacting with Hispanic/Spanish-speaking animal caretakers were investigated through four different answer choices. According to the results, 46% of respondents agreed that it is difficult for them to understand what the caretakers tell them (CI: 30–62%), and 18% strongly agreed (CI: 21–53%) with this statement. Additionally, 36% of respondents agreed (CI: 21–52%) and 31% strongly agreed that they have a challenging time expressing their thoughts in words in the Spanish language (CI: 17.5–47.7%). Furthermore, 21% of respondents agreed (CI: 9–36%) and 31% strongly agreed (CI: 17–48%) that it is difficult for them to understand what is written in Spanish. Lastly, 23% of respondents agreed (CI: 11–40%) and 41% strongly agreed (CI: 25–57%) that they do not feel confident writing things in Spanish. These results indicate that the professionals face greater difficulties in comprehending verbal communication and expressing their ideas orally than understanding and producing written Spanish (Table 1). Based on these results, communication difficulties in the oral medium for both oral receptive skills (listening) and oral productive skills (speaking) are the major barriers that animal professionals need to overcome in their attempt to communicate with the Hispanic workforce on farm settings.

#### 3.3.2. The Need for Spanish in Farm Activities

The necessity of animal professionals to understand, speak, read materials, and write information in Spanish was evaluated. Notably, most participants agreed (39%; CI: 24–56%) and strongly agreed (55%; CI: 38–71%) with the main need to understand what other people are saying. Similarly, 32% (CI: 18–48%) of respondents expressed agreement and 62% (CI: 44–75%) strongly agreed that speaking is a factor of importance regarding the need for using this oral productive skill when interacting with Hispanic/Spanish-speaking animal caretakers. On the other hand, a lower percentage of the animal professionals agreed (32% CI: 18–48%) and strongly agreed (29% CI: 15–46%) that reading materials in Spanish is relevant for carrying out on-farm tasks. Another proportion of respondents considered that writing information in Spanish is relevant for Hispanic/Spanish-speaking animal caretakers to comply with job responsibilities (42%, CI: 26–59% agreed, and 37%, CI: 22–54% strongly agreed). These results provide insights into the self-perceived needs of language skills that professionals deem relevant to their ability to communicate effectively in Spanish with farm Hispanic/Spanish-speaking animal caretakers (Table 2).

#### 3.3.3. Agriculture Professionals’ Purposes for Communicating Orally with Hispanic/Spanish-Speaking Animal Caretakers

The main purposes of animal professionals in terms of oral communication with Hispanic/Spanish-speaking animal caretakers were evaluated. The survey included 13 answer choices for this subitem. Most of the participants expressed agreement in specific key areas: advising on treatment administration (64% strongly agree CI: 47.1–78.3%), explaining animal management protocols (64% CI: 47.1–78.3%), understanding descriptions about animal conditions (59% strongly agree CI: 42.4–73.8%), and instructing on humane handling and restraint (59% strongly agree CI: 44.6–76.1%) (Table 3).

Additionally, the participants agreed that Spanish is relevant for understanding: animal behavioral changes (59% strongly agree CI: 42.4–73.8%), oral explanations about diseases symptoms (56% CI: 39.7–71.8%) and for discussing record keeping (51% strongly agree, CI: 35%.0–67.2%). These findings highlight the self-assessed purpose for using Spanish to communicate with Hispanic/Spanish-speaking animal caretakers while performing husbandry practices.

#### 3.3.4. Agriculture Professionals’ Purposes for Communicating in Writing with Hispanic/Spanish-Speaking Animal Caretakers

The perceived importance of animal professionals for acquiring specific written Spanish language skills for the purpose of engaging with caretakers in farm activities was evaluated. Out of the total, 38% (CI: 24.4–56.5%) of the respondents agreed and 26% strongly agreed (CI: 13.9–43.3%) on the importance of understanding animal records written by caretakers. The same percentage of respondents expressed strong agreement (38% CI: 24.4–56.5%) and agreement (38% CI: 24.4–56.5%) with the importance of providing written instructions about treatments. For skills related to feeding programs, 38% (CI: 24.4–56.5%) agreed and 28% (CI: 15.9–46.1%) strongly agreed that this was an important area for using Spanish. As for the preparation of supplemental feed, 38% (CI: 24.4–56.5%) agreed and 23% (CI: 12.0–40.6%) strongly agreed on this being another purpose for communicating in written Spanish. Respondents’ answers to the item on using written guidelines for animal genetic improvement and breeding programs were 26% (CI: 13.9–43.3%) for strong agreement and 23% (CI: 12.0–40.6%) for agreement. Additionally, participants emphasized the importance of providing written advice on animal welfare (44% agreement CI: 28.9–61.5%; 38% strong agreement CI: 24.4–56.5%). They also emphasized the importance of written explanations about animal inspection (36% agreement CI: 22.2–54.0% and 28% strong agreement CI: 15.9–46.1%). The item written protocols on animal husbandry practices showed 38% (CI: 24.4–56.5%) agreement and 31% (CI: 26.7–59.0%) strong agreement. These findings offer valuable insights into the perceived importance that animal professionals have in relation to the purpose for using written skills in Spanish in the context of on-farm activities, particularly in a Spanish-speaking environment (Table 4).

#### 3.3.5. The Main Aspects of the Spanish Language Necessary for Interacting with Spanish-Speaking Animal Caretakers

This section of the survey established the essential aspects of the Spanish language needed when interacting with Hispanic/Spanish-speaking animal caretakers in the following categories: grammar, agricultural vocabulary, listening comprehension, speaking, reading, and writing. Regarding grammar, 32% of the participants (CI:18–48.7%) agreed and 11% strongly agreed (CI: 3–25.7%) that robust grammar skills were crucial. Agriculture vocabulary presents a 36% agreement (21.6–52.8%) and a substantial 56% of strong agreement (CI: 39.7–71.8%), suggesting a perceived importance of specific vocabulary in farm contexts. In terms of listening comprehension, 49% (32.7–64.9%) agreed and 46% strongly agreed (CI: 30.4–62.6%) on the importance of this language skill, while 44% (28.1–60.2%) of respondents agreed and 49% (32.7–64.9%) strongly agreed on the necessity of mastering speaking skills. For reading, participants emphasized its relevance by expressing 50% (CI: 33.6–66.3%) agreement and 13% (4.9–28.8%) strong agreement. Writing skills also garnered importance, with 49% of the surveyed participants expressing agreement (32.7–64.9%) and 15% (6.0–31.2%) manifesting strong agreement. These findings suggest that a well-rounded language proficiency, including grammar, vocabulary, and the four language macro-skills, are crucial for effective interaction in this specific setting. Nevertheless, it is important to highlight that oral skills (listening and speaking) showed a greater relevance for communicating with Hispanic/Spanish-speaking animal caretakers than written skills (reading and writing). This becomes evident by comparing the agreement and strong agreement scores that listening and speaking received (95% and 93%) against the agreement and strong agreement scores assigned to reading and writing (63% and 64%). The survey highlights specific areas where animal professionals need to focus their attention on to improve communication with the Hispanic/Spanish-speaking animal caretakers. (Table 5)

#### 3.3.6. Topics That Are Likely to Be Addressed When Animal Professionals Communicate with Hisapanic/Spanish-Speaking Caretakers

These sections explored respondents’ perceptions on the topics that are likely to be addressed when animal professionals communicate with Hispanic/Spanish-speaking animal caretakers. The data revealed that, across the different sub-items, most animal professionals strongly agreed or agreed with each of the answer choices of the 23 proposed topics. The following were the most notable responses indicating that these topics are relevant in on-farm settings. Typical clinical signs of illness: 67% strongly agreed (CI: 49.6–80.4%) and 31% (17.5–47.7%) agreed; Euthanasia: 64% strongly agreed (CI: 47.1–78.3%) and 31% agreed (CI: 17.5–47.7%); Treatment—Drugs: 64% strongly agreed (CI: 47.1–78.3%) and 28% agreed (CI: 15.5–45.1%); Identification of sick or injured individuals: 69% strongly agreed (CI: 52.2–82.4%) and 23% agreed (CI: 11.7–39.7%); Animal behavior and emotional states: 51% strongly agreed (CI: 35–67.2%) and agreed 41% (CI: 25.9–57.8%); Biosecurity: 54% strongly agreed (CI: 37.3–69.5%) and 38% agreed (CI: 23.8–55.3%); Body condition: 53% strongly agreed (CI: 36.0–68.6%) and 39% agreed (CI: 24.4–56.5%) (Table 6).

Overall, these results highlight the varying degrees of emphasis that animal professionals place on the topics that need to be dealt with during their interactions with Hispanic/Spanish-speaking animal caretakers. Consequently, training programs that seek to develop communication skills in Spanish should take into consideration the topics that have been identified as important when animal professionals communicate with Hispanic/Spanish-speaking animal caretakers.

## 4. Discussion

The findings of the current study underscore the complex nature of the language barriers that animal professionals face in their attempt to interact with Hispanic/Spanish-speaking animal caretakers in farm settings [41]. The demographic section of the survey provides a comprehensive overview of the sample, revealing a predominantly male and white cohort with diverse educational backgrounds and professional roles, further enriching the dataset. The study emphasizes the challenges in cross-cultural communication, as a considerable number of respondents reported difficulties in understanding verbal communication, articulating thoughts in words, interpreting written Spanish, and expressing confidence in written communication in Spanish.

The examination of language needs among participants reveals a complex array of self-perceived language barriers, pointing to a concerning declining confidence and proficiency in both oral and written Spanish communication. This situation is particularly important when we consider its implications within professional contexts, where effective communication is essential for the completion of tasks ranging from providing advice on treatment administration to explaining animal management protocols [42]. This perceived decline in language confidence and proficiency among animal professionals may impede their ability to engage meaningfully with Hispanic/Spanish-speaking animal caretakers, and may potentially compromise the welfare of the animals under their care. These findings are similar to those reported by Clouser et al. [43] where the inability of the supervisor to speak Spanish was related to an increased risk of occupational injury. The identification of a potential disadvantage for non-bilingual professionals compared to bilingual counterparts adds a valuable layer of insight [44].

Furthermore, the analyzed data across various demographic variables provide a better understanding of how this increases the gap in communication. Notably, non-bilingual professionals, when compared to their bilingual counterparts, showed statistically significant odds or tendencies towards strong agreement on difficulties in understanding and expressing themselves in Spanish. Based on the results, communication difficulties in the oral medium for both oral receptive skills (listening) and oral productive skills (speaking) are the major barriers that animal professionals need to overcome in their attempt to communicate with the Hispanic workforce.

Drawing parallels from Divi et al.’s [45] pilot study on language proficiency and adverse events in U.S. hospitals, the potential consequences of language barriers in professional settings extend beyond healthcare to the agricultural sector. Their findings stress the critical impact of language proficiency on communication in healthcare, revealing a correlation between limited language skills and adverse events [45]. Transposing these insights to the context of agriculture, where animal professionals interact with Hispanic/Spanish-speaking animal caretakers, poses a similar concern.

Notable gender-based differences emerged, with female animal professionals more likely to emphasize the crucial role of specific elements of the Spanish language, such as agricultural terminology, listening comprehension, and speaking abilities. These results resemble the ones reported by Rees and Sheard [46], where female participants were having more positive attitudes towards learning communication skills compared with male participants highlighting a distinctive, gender-related viewpoint on the language skills necessary for efficient communication in animal care settings.

Furthermore, differences related to having a Hispanic background were evident, emphasizing the importance of written language skills for non-Hispanic animal professionals engaged in on-farm activities. These findings support the need for tailored language training programs of Spanish for Specific Purposes in Agriculture that consider the diverse linguistic backgrounds of professionals in the agricultural sector. The development of language courses focused on agriculture with industry-specific vocabulary and communication scenarios is vitally important. Addressing language barriers is crucial for effective communication and for the health and welfare of animals under the care of Hispanic/Spanish-speaking animal caretakers. Overcoming such barriers also contributes to fostering a more inclusive work environment and to enhance the overall performance of professionals in the farming industry [10]. Additionally, this paper provides insights into the self-perceived needs of language skills that animal professionals deem relevant to their ability to communicate effectively in Spanish with Hispanic/Spanish-speaking animal caretakers. This survey emphasizes specific areas where animal professionals need to focus their attention on to improve communication with Hispanic/Spanish-speaking animasl caretakers. The results reveal a need to devote additional efforts to enhance listening and speaking skills.

Overall, these results highlight the varying degrees of emphasis that animal professionals place on different aspects of communication with Hispanic/Spanish-speaking animal caretakers in the areas of animal health, welfare, and animal management. This sheds light on a potential disadvantage experienced by non-bilingual professionals in their interactions with Hispanic/Spanish-speaking animal caretakers [23].

In terms of communication challenges, our results indicate that participants face challenges in oral communication with Hispanic/Spanish-speaking animal caretakers. Most of the respondents struggle to understand spoken Spanish and reported deficiencies in speaking and listening skills. This emphasizes the importance of addressing oral communication barriers to facilitate effective interaction on farms. Also, the participants acknowledge the importance of language skills in their interactions with Hispanic/Spanish speaking animal caretakers, with a high percentage of respondents strongly agreeing on the significance of Spanish speaking skills.

The importance of Spanish communication was reinforced with the animal professionals expressing a need for Spanish language oral proficiency mainly to advise on treatment administration, explaining animal management protocols, understanding descriptions of animal conditions, giving instructions on animal handling, and offering advice on animal welfare. Grammar and agricultural vocabulary are recognized as important, emphasizing the necessity of a strong foundation in language structure and specialized terminology. Also, listening comprehension and speaking skills were highlighted, suggesting that oral communication skills are essential for effectively interacting with the Hispanic workforce within this agricultural context.

These results indicate a higher perceived importance of these oral skills compared to reading and writing. This phenomenon directs attention to specific areas where animal professionals can invest efforts to improve their ability to communicate orally with Hispanic/Spanish-speaking animal caretakers. Based on the survey findings, there are clear implications for training and education programs for animal professionals. Tailoring language courses to emphasize oral communication skills, along with providing specialized vocabulary relevant to farm activities could enhance animal professionals’ ability to interact with Hispanic/Spanish-speaking caretakers. Addressing these needs through targeted language training and skill development can contribute to improved communication and collaboration in a multicultural agricultural setting.

## 5. Conclusions

In sum, the results reveal significant communication challenges faced by animal professionals when interacting with Hispanic/Spanish-speaking animal caretakers on farm settings. The respondents consistently report difficulties in oral communication, particularly in understanding verbal communication and lacking both speaking and listening skills. This highlights the urgent need to address oral communication barriers, as effective interaction on-farm is heavily dependent on clear and efficient spoken communication. Therefore, caution should be exercised when generalizing these results to the entire population of animal professionals across the US. Recognizing and addressing these challenges will not only enhance the animal professionals’ ability to convey information, but also promote better understanding and collaboration within multicultural agricultural settings.

The participants emphasized the paramount importance of language skills in their interactions, with a particular focus on speaking. There was a remarkable consensus among the respondents regarding the significance of speaking skills, and to a lesser degree, the importance of reading and writing skills, which highlights the multifaceted nature of language requirements in this context. This insight provides a clear roadmap for professionals seeking to enhance their communication abilities in Spanish-speaking environments. By investing efforts in improving listening and speaking skills, agriculture professionals can bridge communication gaps more effectively. The proactive development and integration of Spanish for Specific Purposes in Agriculture (SSPA) courses, as highlighted in this study, represent a necessary step towards bridging the communication gap. These courses, tailored to the unique needs of veterinary, animal science, and other animal professionals, hold the potential to enhance understanding, collaboration, and efficiency in diverse working environments. It is of vital importance to invest in SSPA programs to ensure that professionals in the agricultural industry are equipped with the necessary language skills to communicate effectively and promote improved animal welfare practices.

This study has illuminated the critical linguistic barriers that exist between English-speaking animal professionals and Hispanic/Spanish-speaking animal caretakers in the U.S. The findings highlight the necessity of effective communication in ensuring optimal animal welfare and the smooth operation of agricultural settings. A significant discovery was the notable difference in communication challenges faced by non-bilingual and bilingual professionals, with the former group experiencing greater difficulties in understanding and expressing themselves in Spanish.

The implications of these findings for training and education programs are evident. Tailoring language courses to focus on oral communication skills and providing specialized vocabulary relevant to farm activities will empower animal professionals to communicate more proficiently with Hispanic/Spanish-speaking animal caretakers, ultimately contributing to improved collaboration and communication in the diverse and multicultural landscape of agriculture. The implications extend beyond mere linguistic competence, reaching into the realms of animal welfare in multicultural farm settings.

Additionally, this study revealed gender-based nuances in language needs, with female professionals emphasizing different aspects of the Spanish language as being crucial for effective communication. This suggests that gender plays a role in shaping the perception and application of language skills in professional scenarios. Moreover, the data pointed to specific areas where animal professionals, particularly those who are non-bilingual, need to improve their Spanish language skills for more effective on-farm communication.

While this study offers valuable insights, it also acknowledges its limitations such as sample size and demographic representation, suggesting the need for more inclusive and comprehensive research in the future. Longitudinal studies and a broader participant base could provide a more detailed understanding of the evolving communication needs in this sector. In conclusion, the communication challenges identified in this study highlight an urgent need for targeted language training and cultural competency development among animal professionals. Addressing these needs will not only improve animal welfare and farm operations, but also contribute to building more inclusive and effective multicultural work environments in the agricultural sector.

## Figures and Tables

**Figure 1 animals-14-00624-f001:**
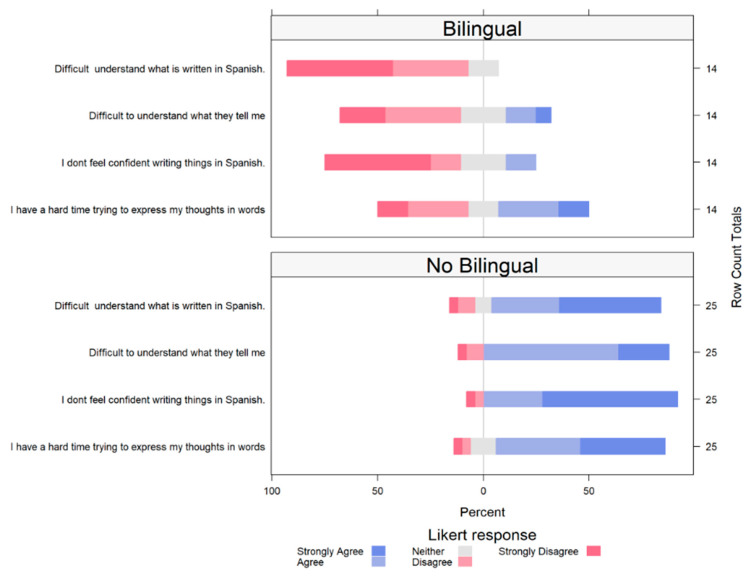
Major issues faced when trying to interact with Hispanic/Spanish-speaking animal caretakers. Comparisons between bilingual and non-bilingual professionals.

**Figure 2 animals-14-00624-f002:**
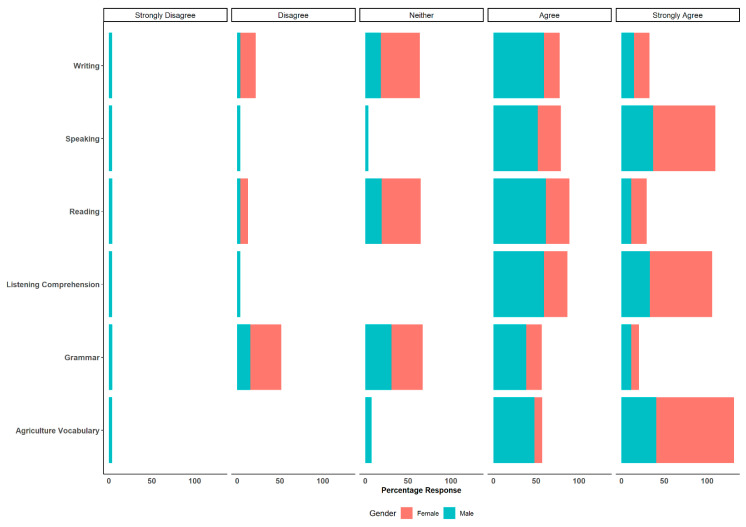
Main aspects of the Spanish language necessary for interacting with Hispanic/Spanish-speaking animal caretakers. Comparisons between female and male professionals.

**Figure 3 animals-14-00624-f003:**
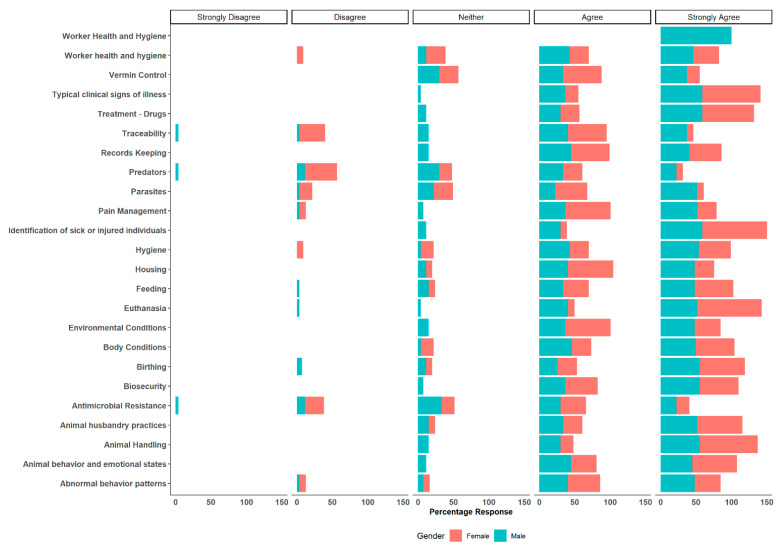
Topics necessary to be addressed when animal professionals communicate with Hispanic/Spanish-speaking animal caretakers. Comparisons between female and male professionals.

**Figure 4 animals-14-00624-f004:**
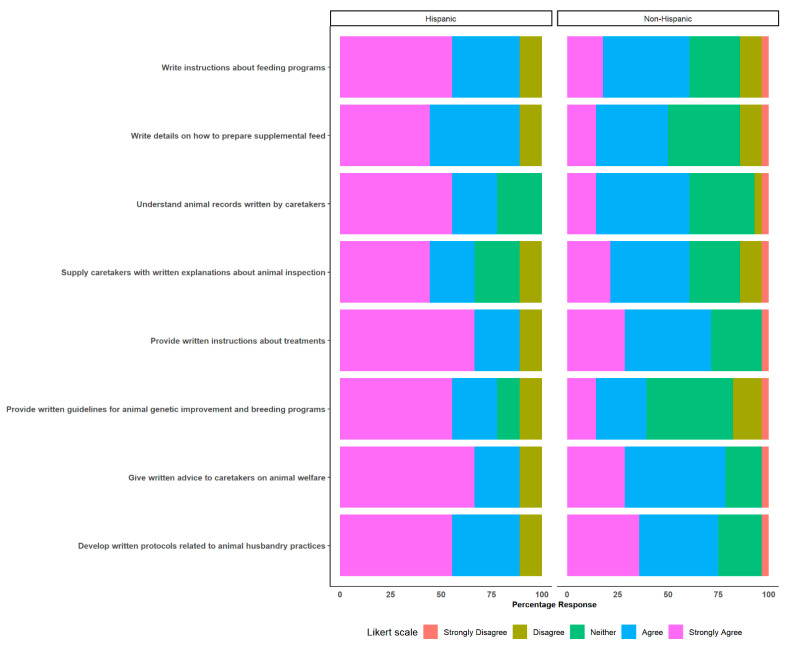
Importance of communicating in written language (Spanish) for the development of on-farm activities, Comparisons between Hispanic and non-Hispanic professionals.

**Figure 5 animals-14-00624-f005:**
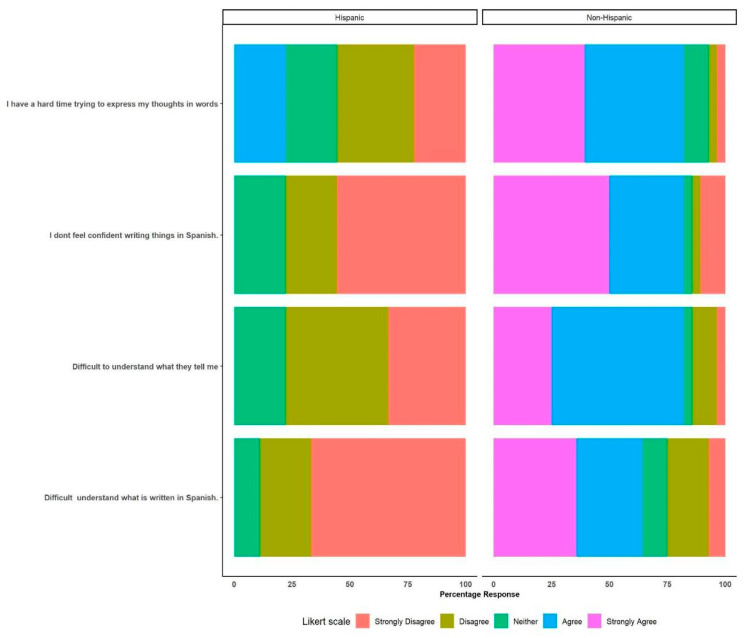
Major issues encountered when interacting with Hispanic/Spanish-speaking animal caretakers. Comparisons between Hispanic and non-Hispanic professionals.

**Figure 6 animals-14-00624-f006:**
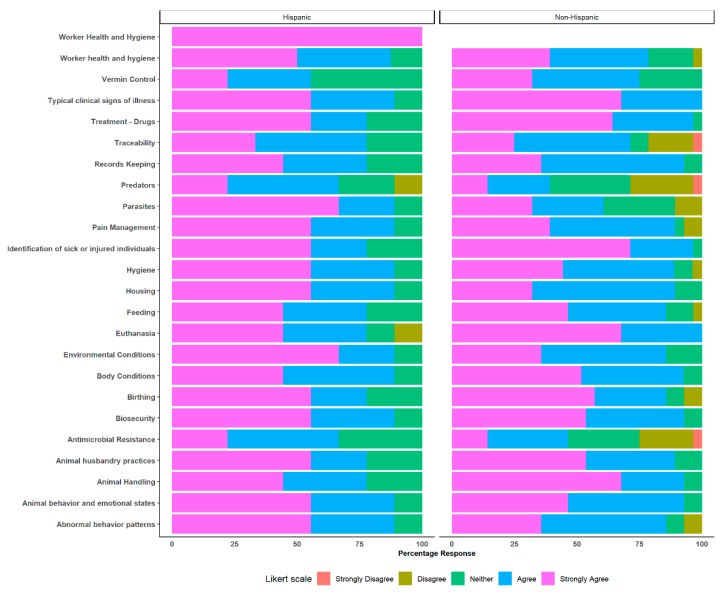
Topics likely to be addressed when animal professionals communicate with Hispanic/Spanish-speaking animal caretakers. Comparisons between Hispanic and non-Hispanic professionals.

**Table 1 animals-14-00624-t001:** Major issues faced when trying to interact with Hispanic/Spanish-speaking animal caretakers on farm.

Answer Choice	Response	Frequency	*n*	Percentage	95% CI
It is difficult for me to understand what they tell me.	1	4	39	10	0.3–25.1
	2	7		18	8.1–34.1
	3	3		8	9.8–36.9
	4	18		46	30.4–62.6
	5	7		18	8.1–34.1
I have a hard time trying to express my thoughts in words.	1	3		8	2–21.9
	2	5		13	4.8–28.2
	3	5		13	4.8–28.2
	4	14		36	21.6–52.8
	5	12		31	17.5–47.7
It is difficult for me to understand what is written in Spanish.	1	8		21	9.8–36.9
	2	7		18	8.1–34.1
	3	4		10	3.3–25.1
	4	8		21	9.8–36.9
	5	12		31	17.5–47.7
I don’t feel confident writing things in Spanish.	1	8		21	9.8–36.9
	2	3		8	2–21.9
	3	3		8	2–21.9
	4	9		23	11.7–39.7
	5	16		41	25.9–57.8

Response: 1: Strongly Disagree, 2: Disagree, 3: Neutral, 4: Agree, and 5: Strongly Agree.

**Table 2 animals-14-00624-t002:** Significant issues faced when trying to interact with Hispanic/Spanish-speaking animal caretakers on farm.

Answer Choice	Response	Frequency	*n*	Percentage	(95% CI)
Understand what Spanish-speaking animal caretakers say.	1	1	38	3	1.3–15.4
	2	1		3	1.3–15.4
	3	0		0	0–11.4
	4	15		39	24.4–56.5
	5	21		55	38.4–71
Speak to caretakers.	1	0	38	0	0–11.4
	2	3		7	0.9–19
	3	0		0	0–11.4
	4	12		32	18.0–48.7
	5	23		62	44.2–74.8
Read materials written in Spanish.	1	1	38	3	0.13–15.4
	2	4		11	3.4–25.7
	3	10		26	13.9–43.3
	4	12		32	18.0–48.7
	5	11		29	15.9–46.1
Write information in Spanish that caretakers need for their job.	1	1	38	3	0.13–15.4
	2	2		5	0.9–19.0
	3	5		13	4–28.8
	4	16		42	26.7–59.0
	5	14		37	22.9–54.0

Response: 1: Strongly Disagree, 2: Disagree, 3: Neutral, 4: Agree, and 5: Strongly Agree.

**Table 3 animals-14-00624-t003:** Animal professionals’ purposes for communicating orally with Hispanic/Spanish-speaking animal caretakers.

Answer Choice	Response	Frequency	*n*	Percentage	(95% CI)
Advise caretakers on how to administer treatments to the animals.	1	1	39	2.5	1–15.0
	2	1		2.5	1–15.0
	3	1		2.5	1–15.0
	4	11		28	15.5–45.1
	5	25		64	47.1–78.3
Explain animal management protocols to caretakers.	1	1	39	3	0.1–15.0
	2	1		3	1–15.0
	3	2		5	0.8–18.6
	4	10		26	13.6–42.4
	5	25		64	47.1–78.3
Teach caretakers how to identify compromised animals and timely euthanasia.	1	1	39	3	1–15.0
	2	1		3	1–15.0
	3	2		5	0.8–18.6
	4	10		26	13.6–42.4
	5	25		64	47.1–78.3
Understand what caretakers are describing about the animals.	1	1	39	3	0.13–15.0
	2	1		3	0.13–15.0
	3	1		3	0.13–15.0
	4	13		33	19.5–50.3
	5	23		59	42.4–73.8
Teach caretakers about humane handling and restraint.	1	1	39	3	0–15.0
	2	1		3	0–15.0
	3	2		5	0.89–18.6
	4	11		31	1.55–45.1
	5	24		59	44.6–76.1
Ask caretakers about animal behavioral changes.	1	1	39	3	0.13–15.0
	2	1		3	0.13–15.0
	3	2		5	0.89–18.6
	4	12		31	17.5–47.7
	5	23		59	42.4–73.8
Understand oral explanations about diseases symptoms.	1	1	39	3	0.1–15.0
	2	1		3	0.1–15.0
	3	2		5	0.89–18.6
	4	13		33	19.5–50.3
	5	22		56	39.7–71.8
Give caretakers instructions about husbandry practices.	1	1	39	3	0.1–15.0
	2	2		5	0.89–18.6
	3	0		0	0–11.1
	4	14		36	21.6–52.8
	5	22		56	39.7–71.8
Understand follow-up reports on animal health progress.	1	1	39	3	0.1–15.07
	2	1		3	0.1–15.07
	3	4		10	3.3–25.1
	4	13		33	19.5–50.3
	5	20		51	35.0–67.2
Talk about record keeping.	1	2	39	5	0.89–18.6
	2	2		5	0.89–18.6
	3	4		10	3.3–25.1
	4	11		28	15.5–45.1
	5	20		51	35.0–67.2
Provide instruction on feeding techniques and the preparation of rations.	1	2	39	5	0.89–18.6
	2	2		5	0.89–18.6
	3	3		8	2–21.9
	4	13		33	19.5–50.3
	5	19		49	32.7–64.9
Inform caretakers about animal inspection protocols.	1	1	39	3	0.01–15.0
	2	5		13	4.81–28.2
	3	1		3	0.01–15.0
	4	14		36	21.6–52.8
	5	18		46	30.4–62.6
Understand reports on animal growth and development.	1	1	39	2	0.1–15.07
	2	3		8	2–21.9
	3	6		15	6.41–31.2
	4	13		33	19.5–50.3
	5	16		41	25.9–57.8

Response: 1: Strongly Disagree, 2: Disagree, 3: Neutral, 4: Agree, and 5: Strongly Agree

**Table 4 animals-14-00624-t004:** Animal professionals’ purposes for communicating in writing with Hispanic/Spanish-speaking animal caretakers.

Answer Choice	Response	Frequency	*n*	Percentage	(95% CI)
Understand animal records written by caretakers.	1	1	39	3	0.1–15.4
	2	1		3	0.1–15.4
	3	12		31	18.0–48.7
	4	15		38	24.4–56.5
	5	10		26	13.9–43.3
Provide written instructions about treatments.	1	1	39	3	0.1–15.4
	2	1		3	0.1–15.4
	3	7		18	8.3–34.8
	4	15		38	24.4–56.5
	5	15		38	24.4–56.5
Write instructions about feeding programs.	1	1	39	3	0.1–15.4
	2	4		10	3.42–25.7
	3	8		21	10.1–37.7
	4	15		38	24.4–56.5
	5	11		28	15.9–46.1
Write details on how to prepare supplemental feed.	1	1	39	3	0.1–15.4
	2	4		10	3.42–25.7
	3	10		26	13.9–43.3
	4	15		38	24.4–56.5
	5	9		23	12.0–40.6
Provide written guidelines for animal genetic improvement and breeding programs.	1	1	39	3	0.1–15.4
	2	5		13	4.9–28.8
	3	14		36	22.2–54.0
	4	9		23	12.0–40.6
	5	10		26	13.9–43.3
Give written advice to caretakers on animal welfare.	1	1	39	3	0.01–15.4
	2	1		3	0.01–15.4
	3	5		13	4.9–28.8
	4	17		44	28.9–61.5
	5	15		38	24.4–56.5
Supply caretakers with written explanations about animal inspection.	1	1	39	3	0.01–15.4
	2	4		10	3.42–25.7
	3	9		23	12.0–40.6
	4	14		36	22.2–54.0
	5	11		28	15.9–46.1
Develop written protocols related to animal husbandry practices.	1	1	39	3	0.1–15.4
	2	1		3	0.1–15.4
	3	6		15	6–31.9
	4	15		38	24.4–56.5
	5	16		41	26.7–59.0

Response: 1: Strongly Disagree, 2: Disagree, 3: Neutral, 4: Agree, and 5: Strongly Agree.

**Table 5 animals-14-00624-t005:** Main aspects of the Spanish language necessary for interacting with Spanish-speaking animal caretakers.

Answer Choice	Response	Frequency	*n*	Percentage	(95% CI)
Grammar	1	1	38	2	0.1–15.4
	2	8		21	10.1–37.7
	3	13		34	20.1–51.4
	4	12		32	18.0–48.7
	5	4		11	3–25.7
Agriculture Vocabulary	1	1	39	3	1.3–15.0
	2	0		0	0–11.1
	3	2		5	0.8–18.6
	4	14		36	21.6–52.8
	5	22		56	39.7–71.8
Listening Comprehension	1	1	39	3	0.1–15.4
	2	1		3	0.1–15.4
	3	0		0	0–11.1
	4	19		49	32.7–64.9
	5	18		45	30.4–62.6
Speaking	1	1	39	3	0.1–15.4
	2	1		3	0.1–15.4
	3	1		3	0.1–15.4
	4	17		43	28.1–60.2
	5	19		48	32.7–64.9
Reading	1	1	38	3	1.3–15.4
	2	2		5	0.9–19
	3	11		29	15.9–46.1
	4	19		50	33.6–66.3
	5	5		13	4.9–28.8
Writing	1	1	39	3	0.1–15
	2	3		8	2–21.9
	3	10		26	13.6–42.4
	4	19		48	32.7–64.9
	5	6		15	6–31.2

Response: 1: Strongly Disagree, 2: Disagree, 3: Neutral, 4: Agree, and 5: Strongly Agree.

**Table 6 animals-14-00624-t006:** Topics that are likely to be addressed when animal professionals communicate with caretakers.

Answer Choice	Response	Frequency	*n*	Percentage	(95% CI)
Environmental conditions	1	0	39	0	0–11.1
	2	0		0	0–11.1
	3	5		13	4–28.2
	4	17		44	28.1–60.2
	5	17		44	28.1–60.2
Animal behavior and emotional states	1	0	39	0	0–11.1
	2	0		0	0–11.1
	3	3		8	2–21.9
	4	16		41	25.9–57.8
	5	20		51	35–67.2
Identification of sick or injured individuals	1	0	39	0	0–11.1
	2	0		0	0–11.1
	3	3		8	2–21.9
	4	9		23	11.7–39.7
	5	27		69	52.2–82.4
Typical clinical signs of illness	1	0	39	0	0–11.1
	2	0		0	0–11.1
	3	1		3	0.1–15.07
	4	12		31	17.5–47.7
	5	26		67	49.6–80.4
Abnormal behavior patterns	1	0	39	0	0–11.1
	2	2		5	0.8–18.6
	3	3		8	2–21.9
	4	17		44	28.1–60.2
	5	17		44	28.1–60.2
Pain management	1	0	39	0	0–11.1
	2	2		5	0.8–18.6
	3	2		5	0.8–18.6
	4	17		44	28.1–60.2
	5	18		46	30.4–62.6
Treatment—Drugs	1	0	39	0	0–11.1
	2	0		0	0–11.1
	3	3		8	2–21.9
	4	11		28	15.5–45.1
	5	25		64	47.1–78.3
Parasites	1	0	39	0	0–11.1
	2	3		8	2–21.9
	3	9		23	11.7–39.7
	4	11		28	15.5–45.1
	5	16		41	25.9–57.8
Vermin control	1	0	39	0	0–11.1
	2	0		0	0–11.1
	3	11		28	15.5–45.1
	4	16		41	25.9–57.8
	5	12		31	17.5–47.7
Biosecurity	1	0	39	0	0–11.1
	2	0		0	0–11.1
	3	3		8	2–21.9
	4	15		38	23.8–55.3
	5	21		54	37.3–69.5
Hygiene	1	0	38	0	0–11.4
	2	1		3	0.1–15.4
	3	3		8	2.0–22.4
	4	15		39	24.4–56.5
	5	19		50	34.8–65.1
Animal husbandry practices	1	0	39	0	0–11.1
	2	0		0	0–11.1
	3	5		13	4.8–28.2
	4	12		31	17.5–47.7
	5	22		56	39.7–71.8
Body conditions	1	0	38	0	0–11.4
	2	0		0	0–11.4
	3	3		8	2.0–22.4
	4	15		39	24.4–56.5
	5	20		53	36.0–68.6
Feeding	1	0	39	0	0–11.1
	2	1		3	0.13–15.0
	3	5		13	4.8–28.2
	4	14		36	21.6–52.8
	5	19		49	32.7–64.9
Housing	1	0	39	0	0–11.1
	2	0		0	0–11.1
	3	4		10	3.3–25.1
	4	19		49	32.7–64.9
	5	16		41	25.9–57.8
Predators	1	1	39	3	0.13–15.0
	2	8		21	9.8–36.9
	3	11		28	15.5–45.1
	4	12		31	17.5–47.7
	5	7		18	8.1–34.1
Birthing	1	0	39	0	0–11.1
	2	2		5	0.89–18.6
	3	4		10	3–25.1
	4	10		26	13.6–42.4
	5	23		59	42.1–74.0
Records keeping	1	0	39	0	0–11.1
	2	0		0	0–11.1
	3	4		10	3–25.1
	4	19		49	32.7–64.9
	5	16		41	25.9–57.8
Euthanasia	1	0	39	0	0–11.1
	2	1		3	0.13–15.0
	3	1		3	0.13–15.0
	4	12		31	17.5–47.7
	5	25		64	47.1–78.3
Traceability	1	1	39	3	0.13–15.0
	2	5		13	4.87–28.2
	3	4		10	3.3–25.1
	4	17		44	28.1–60.2
	5	12		31	17.5–47.7
Antimicrobial resistance	1	1	39	3	0.13–15.0
	2	6		15	6–31.2
	3	11		28	15.5–45.1
	4	13		33	19.5–50.3
	5	8		21	9.8–36.9
Animal handling	1	0	39	0	0–11.1
	2	0		0	0–11.1
	3	4		10	3.3–25.1
	4	10		26	13.6–42.4
	5	25		64	47.1–78.3
Worker health and hygiene	1	0	39	0	0–11.1
	2	1		3	0.13–15.0
	3	6		15	6.4–31.2
	4	14		36	21.6–52.8
	5	18		46	30.4–62.6

## Data Availability

All data included in the manuscript and supplementary data are available at Dr. Arlene Garcia’s Laboratory.

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
