# Peer review of "Understanding Communication Barriers: Demographic Variables and Language Needs in the Interaction between English-Speaking Animal Professionals and Spanish-Speaking Animal Caretakers"

_animals, 2024, doi:10.3390/ani14040624_

Round 1
Reviewer 1 Report
Comments and Suggestions for Authors
The manuscript provides a very interesting insight into language barriers and critical aspects of communication by veterinary, animal science, and agricultural professionals who interact directly at various levels with Spanish-speaking animal caretakers working in the field. The findings and discussion elements outlined in the study certainly lay the groundwork for a broader and deeper study aimed at improving the language skills of professionals to improve interactions with Spanish-speaking farm animal caretakers and, ultimately, animal health and welfare.
Before proceeding to publication, the reviewer would like to draw attention to several aspects that could be revised to improve accessibility and clarity of content.
General notes: A reading of the manuscript shows the participation in the study of three main professional categories: veterinarians, animal scientists, and agricultural professionals. Nevertheless, the three categories are not reported uniformly in the text. As evidence, for example, in the title only the category of agricultural professionals is mentioned, in contrast, the category of agricultural professionals does not appear in the scope line in the abstract although later in the text (lines 26-29) specific results on agricultural professionals responding to the questionnaire are shown. Please amend the title to reflect the study and amend the abstract to provide a clear view of the results obtained from all the participants' categories.
The same suggestion should also be applied to the main text, particularly in the experimental part, where the three categories are not always mentioned. A prominent example is found in the scope in which agricultural professionals are not mentioned.
In subsection 2.1 instrument survey, please include a graphic representation of the questionnaire administered.
2.2 "General demographics": unclear subsection whose title does not seem appropriate to the content. If the reviewer's interpretation is correct, the subsection seems to be a description of all the sections of the questionnaire introduced in the previous subsection, and not just the section on general demographics.
A section 1 and section 2 appear in the text which have not been introduced previously. Please rewrite this part to clarify the content.
Specific notes:
Line 66: please specify the term "animal scientists" refers to which professional figures
Line 131: The reviewer would suggest including a definition of needs analysis, even a brief one, in line with West (1994), to improve immediate understanding of the context.
Line 244: Please amend the sentence also including agricultural professionals which are reported in section 3.1 lines 359-360 among respondents.
Line 350-355: The sentence is confusing, first saying that most of the participants belong to two of the three categories, a small percentage to the category of agricultural professionals as defined above, and then 46.15% of the participants who seem not to belong to any of the three categories. This 46.15% would represent the majority, contrary to what was said at the beginning of the sentence. Moreover, regarding this last group, and the types mentioned (animal nutritionists, farm advisors, farm trainers, and professors), could they not be included in the categories proposed?
Lines 778-779: The extension of the critical aspects and barriers to communication with Hispanic workers that emerged from the study, although very interesting and well described, to the whole of the US territory seems a little extreme given the limited number of participants, as the authors also acknowledge.
Note on the text :
Please check the punctuation throughout the text and in particular the use of parentheses, which are sometimes open and not closed, or present as typing errors not been removed (for an example see lines 102, 134).
Graphics: Please improve the readability of graphics by increasing the font size
Author Response
Thank you for taking the time to review our manuscript. We are grateful for your suggestions. We have addressed all of your suggested revisions.
Reviewer 1:
General notes: A reading of the manuscript shows the participation in the study of three main professional categories: veterinarians, animal scientists, and agricultural professionals. Nevertheless, the three categories are not reported uniformly in the text. As evidence, for example, in the title only the category of agricultural professionals is mentioned, in contrast, the category of agricultural professionals does not appear in the scope line in the abstract although later in the text (lines 26-29) specific results on agricultural professionals responding to the questionnaire are shown. Please amend the title to reflect the study and amend the abstract to provide a clear view of the results obtained from all the participants' categories.
- The title and abstract were amended to reflect the participants in all the categories “animal professionals”
The same suggestion should also be applied to the main text, particularly in the experimental part, where the three categories are not always mentioned. A prominent example is found in the scope in which agricultural professionals are not mentioned.
- The changes were made throughout the whole text.
In subsection 2.1 instrument survey, please include a graphic representation of the questionnaire administered.
- A graphical representation of the questionnaire was added in the supplementary data
2.2 "General demographics": unclear subsection whose title does not seem appropriate to the content. If the reviewer's interpretation is correct, the subsection seems to be a description of all the sections of the questionnaire introduced in the previous subsection, and not just the section on general demographics.
- The tittle was changed for to add clarification.
A section 1 and section 2 appear in the text which have not been introduced previously. Please rewrite this part to clarify the content.
- This section was changed to section B I and B II and describe in the text.
Specific notes:
Line 66: please specify the term "animal scientists" refers to which professional figures
- Change made and highlighted.
Line 131: The reviewer would suggest including a definition of needs analysis, even a brief one, in line with West (1994), to improve immediate understanding of the context.
- Definition was added Line 135-146 and highlighted.
Line 244: Please amend the sentence also including agricultural professionals which are reported in section 3.1 lines 359-360 among respondents.
- The profession and occupations were clarified.
Line 350-355: The sentence is confusing, first saying that most of the participants belong to two of the three categories, a small percentage to the category of agricultural professionals as defined above, and then 46.15% of the participants who seem not to belong to any of the three categories. This 46.15% would represent the majority, contrary to what was said at the beginning of the sentence. Moreover, regarding this last group, and the types mentioned (animal nutritionists, farm advisors, farm trainers, and professors), could they not be included in the categories proposed?
- This section was corrected to add clarification.
Lines 778-779: The extension of the critical aspects and barriers to communication with Hispanic workers that emerged from the study, although very interesting and well described, to the whole of the US territory seems a little extreme given the limited number of participants, as the authors also acknowledge.
- We appreciate the reviewer's observation regarding the extension of critical aspects and barriers to communication with Hispanic workers to the entire US territory. We acknowledge the limitation in our study, which is based on a limited number of participants. The decision to generalize our findings was made with caution, considering animal professionals from different backgrounds as veterinarians, animal scientists, farm workers, and farm owners. We understand the concern raised and have added a clarification in the manuscript (Line 832-834) to emphasize the cautious interpretation of our findings beyond the studied sample size.
Note on the text:
Please check the punctuation throughout the text, and particularly the use of parentheses, which are sometimes open and not closed, or present as typing errors not been removed (for an example see lines 102, 134).
- Punctuation throughout the text was checked and typing errors were removed.
Reviewer 2 Report
Comments and Suggestions for Authors
Authors evaluated perceived cross-cultural communication challenges between English and Spanish speaking professionals and workers within the animal agriculture industry. The manuscript is well written and in good condition. Authors describe their methods well, and justify the outcomes. Though the sample size is admittedly small, this manuscript highlights important aspects of equity, justice, and efficacy in animal welfare in aspects that are underrepresented in published literature, and as such, is a valuable contribution to science.
I only have a few detailed thoughts:
Authors should perform a close review for several edits, mainly floating punctuation (e.g., parenthesis in Lns 102 and 134 and comma spacing in Ln 148), possible missing word in Ln 283 causing a sentence fragment (e.g., “…: WHAT topics are…” or “…: topics THAT are…”), and missing letter in Ln 599 (macro-Skills).
Lns 246-248, 259: The content of the survey makes sense, but as written, the section/part headings do not make sense. Is Section A mentioned in Ln 246 the same as Section 1 mentioned in Ln 259? Likewise, are Part BI and BII mentioned on Ln 247 a separate Section? If not, where is Part A? Recommend clarifying survey headings.
Lns 354-355. Given 'animal professionals' was the largest respondent sampling, I recommend authors revise their abstract, Introduction, and discussion to reflect this finding, vs ‘veterinarians’ being the leading occupation throughout much of the manuscript though they constituted less than 20% of respondents here.
Lns 357-360. Again, this statement as written appears to conflate the proportion of respondents with a veterinary degree, though those degrees were less than 20% of respondents. Recommend revising. Likewise, are these degrees exclusive, as in highest degree attained, or did respondents note all degrees they had?
Ln 534 and Ln 555. Are the underlined portions of the headings necessary?
Authors used a good portion of the Introduction discussing Needs Analysis/Assessment, but the topic is not revisited in the Discussion. Though the manuscript is on the long side, I’d recommend consolidating this topic in the Introduction to parallel the Discussion, discussing if the objectives of the Needs Assessment was accomplished through the survey in the Discussion, or some combination of both.
Author Response
Thank you for taking the time to review this manuscript. We greatly appreciate your suggestions. All of your comments have been addressed below and in the manuscript.
Reviewer 2 comments:
- Authors should perform a close review for several edits, mainly floating punctuation (e.g., parenthesis in Lns 102 and 134 and comma spacing in Ln 148), possible missing word in Ln 283 causing a sentence fragment (e.g., “…: WHAT topics are…” or “…: topics THAT are…”), and missing letter in Ln 599 (macro-Skills).
Floating punctuation, and coma spacing has been removed/fixed (Lns 102, 134,148)
Lns 246-248, 259: The content of the survey makes sense, but as written, the section/part headings do not make sense. Is Section A mentioned in Ln 246 the same as Section 1 mentioned in Ln 259? Likewise, are Part BI and BII mentioned on Ln 247 a separate Section? If not, where is Part A? Recommend clarifying survey headings.
- Modified for clarity.
Lns 354-355. Given 'animal professionals' was the largest respondent sampling, I recommend authors revise their abstract, Introduction, and discussion to reflect this finding, vs ‘veterinarians’ being the leading occupation throughout much of the manuscript though they constituted less than 20% of respondents here.
- Changes were made accordingly throughout all the text.
Lns 357-360. Again, this statement as written appears to conflate the proportion of respondents with a veterinary degree, though those degrees were less than 20% of respondents. Recommend revising. Likewise, are these degrees exclusive, as in highest degree attained, or did respondents note all degrees they had?
- We changed the name of the respondents according to animal professionals. The degrees were exclusive. The respondents only had the option to select one academic degree title, so it is assumed that they mentioned their highest attained academic degree.
Ln 534 and Ln 555. Are the underlined portions of the headings necessary?
- We removed the underlined portions of the headings.
Authors used a good portion of the Introduction discussing Needs Analysis/Assessment, but the topic is not revisited in the Discussion. Though the manuscript is on the long side, I’d recommend consolidating this topic in the Introduction to parallel the Discussion, discussing if the objectives of the Needs Assessment was accomplished through the survey in the Discussion, or some combination of both.
- A paragraph was added in the discussion section related with this topic.